# Camera Calibration for Coastal Monitoring Using Available Snapshot Images

**Gonzalo Simarro** [1,*,†] , **Daniel Calvete** [2,†] , **Paola Souto** [3] **and Jorge Guillén** [1]

1   ICM (CSIC), Passeig Marítim de la Barceloneta 37–49, 08003 Barcelona, Spain; jorge@icm.csic.es
2   Universitat Politècnica de Catalunya, Jordi Girona 1–3, 08034 Barcelona, Spain; daniel.calvete@upc.edu
3   Università di Ferrara, Via Saragat 1, 44122 Ferrara, Italy; stcpml@unife.it
*   Correspondence: simarro@icm.csic.es
†   These authors contributed equally to this work.

**Abstract:** Joint intrinsic and extrinsic calibration from a single snapshot is a common requirement in coastal monitoring practice. This work analyzes the influence of different aspects, such as the distribution of Ground Control Points (GCPs) or the image obliquity, on the quality of the calibration for two different mathematical models (one being a simplification of the other). The performance of the two models is assessed using extensive laboratory data (i.e., snapshots of a grid). While both models are able to properly adjust the GCPs, the simpler model gives a better overall performance when the GCPs are not well distributed over the image. Furthermore, the simpler model allows for better recovery of the camera position and orientation.

**Keywords:** coastal video monitoring; camera calibration; sensitivity analysis

## 1. Introduction

Coastal monitoring systems using digital video cameras have become a widely used tool to study near-shore processes since the advent of the ARGUS system over 20 years ago [1,2]. At present, besides the original ARGUS developments, there exists a wide variety of packages to manage image acquisition and processing ([3–6], among others). Video monitoring systems have been shown to be useful, to cite just a few examples, in obtaining intertidal and subaquatic bathymetries [7–9], to detect and analyze shoreline dynamics [10,11], or to study the morphodynamics of beach systems [12,13]. Camera calibration is critical in coastal video monitoring systems, as it allows us to relate pixels in the images to real-world co-ordinates and vice versa.

Camera calibration in coastal video monitoring follows close-range photogrammetric procedures [1,14]. Even though the distance to the objects monitored (i.e., beaches) are up to ∼1000 m, the hypotheses of close-range calibration apply (e.g., no atmospheric refraction or non-negligible lens distortion). Actually, in ARGUS-like fixed stations, it is common practice to obtain the parameters related to lens distortion (intrinsic calibration parameters) prior to final deployment through classic close-range methods, using chessboard or similar patterns [1,6]. The camera position and orientation (extrinsic calibration parameters) are then obtained through Ground Control Points (GCPs); that is, pixels whose real-world co-ordinates are known. The literature on full (intrinsic and extrinsic parameters) close-range camera calibration photogrammetry is extensive, and includes studies on the governing equations [15–17], the calibration procedures [17–21], and applications including structure-from-motion and multi-camera approaches [22–24]. However, there have been few works dealing with the full calibration from a single image using a few GCPs.

In most coastal ARGUS-like monitoring systems, the intrinsic parameters are obtained prior to the final deployment of the camera, as mentioned above, and the extrinsic parameters are then

obtained through GCPs. In many practical situations, however, intrinsic calibration of the camera is not available. This is the case, for example, when using available surfcams around the globe to obtain morphodynamic information [25] or in the CoastSnap project [14]—a citizen science project in which citizens provide smartphone images for some given beaches. In general, taking advantage of the huge amount of freely available coastal images for morphodynamic studies and coastal management is a challenge for the research community. In such situations, all of calibration parameters (i.e., both intrinsic and extrinsic) must be obtained from the GCPs [25]. In a calibration campaign, a large amount of targets (GCPs) can be spread over the entire image and high quality calibrations can be obtained. In the practical situation we want to address, it is only possible to use fixed features and, as large portions of the images are sand, water, or sky, the GCPs are restricted to a relatively small part of the image. For illustrative purposes, Figure 1 includes two snapshots from Castelldefels and Barcelona beaches (Spain, see coo.icm.csic.es): in Figure 1A, the GCPs are usually in the lower half of the image; while in Figure 1B, they mainly lie in the right and lower parts. In addition, the number of points used is usually small; for example, [14] used only seven GCPs for georectification of community-contributed images. Such a low number of GCPs also raises the question of which is—while remaining in the domain of close-range photogrammetry—the most suitable calibration model. Please note that this is very different to what is usually found in close-range photogrammetry, where the calibration is done using a large number of points. In summary, new demands on coastal monitoring systems require further understanding of image calibration when a reduced number of GCPs must be chosen within only a small region of the image.

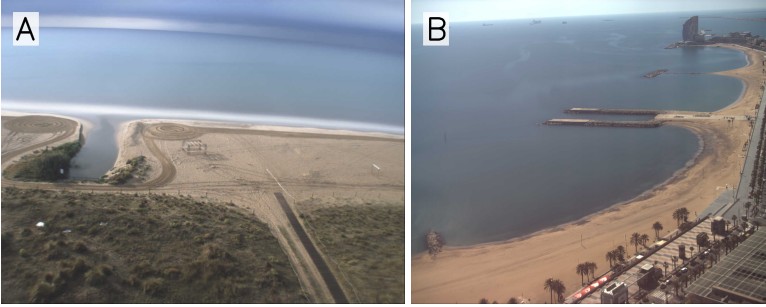

**Figure 1.** Images from Castelldefels (**A**, at 41°15′54.9″ N, 1°59′29.1″ E) and Barcelona (**B**, at 41°23′16.5″ N, 2°11′50.9″ E) video monitoring stations (coo.icm.csic.es).

The main objective of this contribution is to determine the most suitable GCP distributions and calibration model to georectify images on coastal monitoring systems. To do this, we assume that there is only a single snapshot available to obtain a full camera calibration (intrinsic and extrinsic parameters) with a reduced number of GCPs. In addition, the premises of close-range photogrammetry and the non-use of wide angle lenses are considered. Two mathematical models are considered, one being a simplification of the other. The influence of the obliquity of the snapshot or the GCP distribution throughout the image on the calibration quality is analyzed. The ability to accurately recover some useful calibration parameters (e.g., camera position) is also discussed.

## 2. Materials and Methods

### 2.1. Camera Mathematical Models

The pinhole model [26], together with the Brown–Conrady [27] model for decentered lens distortion, are the governing equations typically used for cameras in coastal video monitoring systems; see Figure 2. Given the real-world co-ordinates of a point, $\mathbf{x} = (x, y, z)$, its pixel position, in terms of column $c$ and row $r$, is given by:

$$c = \frac{x_{U\star}\left(1 + k_{1\star}d_{U\star}^2 + k_{2\star}d_{U\star}^4\right) + p_{2\star}\left(d_{U\star}^2 + 2x_{U\star}^2\right) + 2p_{1\star}x_{U\star}y_{U\star}}{s_{c\star}} + o_c, \tag{1a}$$

$$r = \frac{y_{U\star}\left(1 + k_{1\star}d_{U\star}^2 + k_{2\star}d_{U\star}^4\right) + p_{1\star}\left(d_{U\star}^2 + 2y_{U\star}^2\right) + 2p_{2\star}x_{U\star}y_{U\star}}{s_{r\star}} + o_r, \tag{1b}$$

where $k_{1\star}, k_{2\star}, p_{1\star}, p_{2\star}, s_{c\star}, s_{r\star}, o_c$, and $o_r$ are free parameters; higher order distortion terms are avoided for we do not consider wide angle lenses. Furthermore, $d_{U\star}^2 = x_{U\star}^2 + y_{U\star}^2$ and $x_{U\star}$ and $y_{U\star}$ are given by

$$x_{U\star} = -\frac{(\mathbf{x} - \mathbf{x_c}) \cdot \mathbf{e_u}}{(\mathbf{x} - \mathbf{x_c}) \cdot \mathbf{e_f}} + (k_c - o_c)\, s_{c\star}, \tag{2a}$$

$$y_{U\star} = +\frac{(\mathbf{x} - \mathbf{x_c}) \cdot \mathbf{e_v}}{(\mathbf{x} - \mathbf{x_c}) \cdot \mathbf{e_f}} + (k_r - o_r)\, s_{r\star}, \tag{2b}$$

where $\mathbf{x_c} = (x_c, y_c, z_c)$ is the optical center (camera position); $\mathbf{e_u}, \mathbf{e_v}$, and $\mathbf{e_f}$ are orthogonal unit vectors given by the Eulerian angles of the camera (azimuth $\phi$, roll $\sigma$, and tilt $\tau$); and $k_c$ and $k_r$ stand for the pixel co-ordinates of the center of the image (known). The inversion of the above Equations (1) and (2) allows us to obtain the real-world co-ordinates of a pixel if an extra condition is given (typically, the point being in a horizontal plane $z = z_0$); this inversion requires the use of iterative procedures to solve the implicit equations.

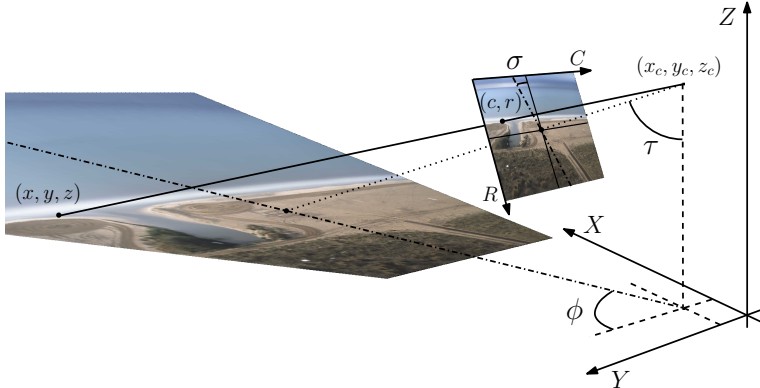

**Figure 2.** Real-world to pixel transformation: camera position $(x_c, y_c, z_c)$ and Eulerian angles ($\phi, \sigma$ and $\tau$).

Overall, 14 parameters need to be established to calibrate the above (mathematical) camera model. The intrinsic parameters are as follows:

- radial and tangential distortions: $k_{1\star}, k_{2\star}, p_{1\star}$, and $p_{2\star}$ (dimensionless);
- pixel size: $s_{c\star}$ and $s_{r\star}$ (dimensionless); and
- decentering: $o_c$ and $o_r$ (in pixels),

and the extrinsic parameters are:

- real world co-ordinates of the center of vision: $x_c, y_c$, and $z_c$ (in units of length); and
- Eulerian angles: $\phi, \sigma$, and $\tau$ (in radians).

The above equations (including the set of 14 parameters) are referred to as the "complete" model, or $M_1$. For most present-day cameras, it is reasonable to assume that the radial distortion is parabolic (i.e., $k_{2\star} = 0$), the tangential distortion is negligible ($p_{1\star} = p_{2\star} = 0$), the pixels are squared ($s_{c\star} = s_{r\star}$), and that the decentering is also negligible ($o_c = k_c$ and $o_r = k_r$). The above hypotheses lead to a "reduced" model, herein called $M_2$, with only 8 free parameters ($x_c, y_c, z_c, \phi, \sigma, \tau, k_{1\star}$, and $s_{c\star}$). Interestingly, the inversion of the model equations becomes explicit when model $M_2$ is considered (i.e., it becomes a cubic equation).

### 2.2. Error Definition and Calibration Procedure

A Ground Control Point (GCP) is a 5-tuple including the real-world co-ordinates of a point and the corresponding pixel co-ordinates (column $c$ and row $r$) in an image (i.e., $(x, y, z, c, r)$). For a set of $n$ GCPs $(x_i, y_i, z_i, c_i, r_i)$ and a camera model with given intrinsic and extrinsic parameters, following [6], the calibration error is defined as

$$\epsilon^* = \sqrt{\frac{1}{n} \sum_{i=1}^{n} \left[ (c_i - c_i^*)^2 + (r_i - r_i^*)^2 \right]}, \tag{3}$$

where $c_i^*$ and $r_i^*$ are the pixel co-ordinates obtained from the corresponding real-world co-ordinates (i.e., $(x_i, y_i, z_i)$) through the camera model for the given parameters. For a certain set of GCPs, an image is here calibrated by finding the parameters (intrinsic and extrinsic) which minimize the above error. The optimization method considered is Broyden–Fletcher–Goldfarb–Shanno (BFGS, [28]) combined with a Monte-Carlo-like seeding method. Usually, the calibration takes only a few CPU seconds.

In real practice, the pixel co-ordinates of GCPs are manually digitized by an expert user, with an unavoidable error that is usually on the order of a few pixels. Understanding the influence of different factors (e.g., the obliquity, the amount and distribution of the GCPs, or the mathematical model) on the propagation of this error to the calibration quality is a key issue. For this reason, $J$ "perturbed" calibrations are performed for each of the analyzed cases in the following section. For each $j$ of these $J$ calibrations, each of the $n$ pixel co-ordinates of the GCPs, originally digitized at $(c_i, r_i)$, was randomly perturbed with a noise of $\pm 2$ pixels (px); that is, $(c_i + \xi_{ij}, r_i + \psi_{ij})$, where $\xi_{ij}$ and $\psi_{ij}$ are realizations of a uniformly distributed random variable in the range $[-2, +2]$. The calibration errors for each of these perturbations are referred to as $\epsilon_P(j)$; that is,

$$\epsilon_P(j) = \sqrt{\frac{1}{n} \sum_{i=1}^{n} \left[ (c_i + \xi_{ij} - c_{ij}^*)^2 + (r_i + \psi_{ij} - r_{ij}^*)^2 \right]}, \tag{4}$$

where $c_{ij}^*$ and $r_{ij}^*$ are the pixel co-ordinates obtained from the corresponding real-world co-ordinates, $(x_i, y_i, z_i)$, for the calibration parameters of the $j$th perturbation. The errors in the real-world GCP co-ordinates are usually negligible in coastal studies (as it is orders of magnitude smaller than the size that the pixel represents in the real world). The errors $\epsilon^*$ and $\epsilon_P$ give a measure of the ability of the camera model to fit the GCPs, either original or perturbed. A different error is introduced below.

Consider $J$ perturbed calibrations and a set of GCPs (here not necessarily those used for the calibration): for each GCP $i$, $(x_i, y_i, z_i, c_i, r_i)$, the error $\tilde{\epsilon}(i)$ is defined as

$$\tilde{\epsilon}(i) = \sqrt{\frac{1}{J} \sum_{j=1}^{J} \left[ (c_i - \tilde{c}_{ij})^2 + (r_i - \tilde{r}_{ij})^2 \right]}, \tag{5}$$

where $(\tilde{c}_{ij}, \tilde{r}_{ij})$ is the pixel co-ordinate obtained from $(x_i, y_i, z_i)$ using the camera mathematical model and the $j$th perturbed calibration parameters. The above error is defined for each pixel of the set of GCPs. The Root Mean Square (RMS) over the set of GCPs is

$$\epsilon_Q = \sqrt{\frac{1}{n} \sum_{i=1}^{n} \tilde{\epsilon}^2(i)}, \tag{6}$$

with $i$ running over all the pixels of the GCPs. The error $\epsilon_Q$ gives a measure of the quality of the calibration for a given set of GCPs. If the GCPs are the same set used to obtain the perturbed calibrations, the error will be referred to as $\epsilon_G$. When there are no pixel perturbations, $\xi_{ij} = \psi_{ij} = 0$ for all $j$ (and $i$) and Equation (4) reduces to Equation (3) (i.e., $\epsilon_P(j) = \epsilon^*$ for all $j$). Furthermore, in the unperturbed case, as the perturbed calibrations become the original unperturbed ones, $\tilde{c}_{ij} = c_i^*$

(for all $j$), such that, from Equation (5), it is $\tilde{\epsilon}^2(i) = |c_i - c_i^*|^2 + |r_i - r_i^*|^2$ and, from Equation (6), $\epsilon_G = \epsilon^* (= \epsilon_P)$.

### 2.3. Experimental Setup

To gain a better understanding of the influence of different aspects on the quality of the calibration and the accuracy of the calibration parameters, a wide range of scenarios was analyzed. Two smartphone cameras were employed: a Samsung Galaxy Grand Prime ($2048 \times 1152$ pixels) and a Xiaomi Redmi 10 ($2016 \times 1512$ pixels). As both cameras gave equivalent results, only one of them (the Samsung) is introduced below, for the sake of clarity. Three different snapshots were taken of the same grid (see Figure 3), in order to consider a range of obliquities (tilt $\tau$): $\tau \sim 55°$ (angle $A_1$), $\tau \sim 40°$ ($A_2$), and $\tau \sim 15°$ ($A_3$, which is almost zenithal). The GCPs were easily obtained in these images as the intersections of the grid lines. The pixel co-ordinates of the GCPs were manually digitized with an error estimated as $\sim$2 px. The unit length in the real-world, "u", was the side of the squares of the grid (around 5 cm).

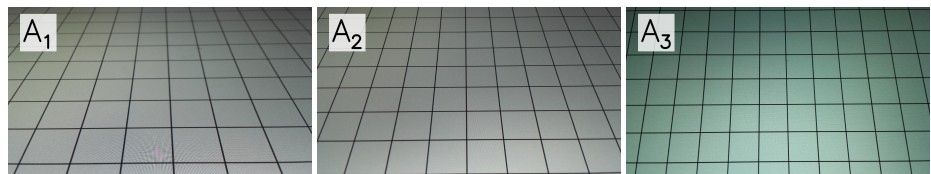

**Figure 3.** Angles $A_1$ ($\tau \sim 55°$), $A_2$ ($\tau \sim 40°$), and $A_3$ ($\tau \sim 15°$) to analyze the influence of obliquity.

For each of the three angles, eight different subsets ($S_0$ to $S_7$) of the whole set of grid intersections ($\sim$80) were considered to be the GCPs for calibration. Figure 4 shows the eight subsets for the angle $A_1$; similar displays were considered for the other images in Figure 3 (although, necessarily, with some differences between the images). While $S_0$ considers all the available intersections of the grid as GCPs, the rest of the sets include eight GCPs distributed in different ways. Leaving aside the especial case $S_0$, some sets correspond to (and are motivated by) real case conditions. For instance, the set $S_3$ resembles Figure 1A and set $S_6$ resembles Figure 1B. The other sets were designed to analyze the results from a more theoretical point of view (e.g., see $S_1$ and $S_2$). The set $S_1$ would correspond to Figure 1B if the horizon line was included (the horizon line is not analyzed in this work). While eight GCPs is a reasonable number of GCPs in usual practice [3,14], and was considered for the reference case, similar displays with 6 and 12 GCPs were also considered for sets $S_1$ to $S_7$.

For each angle and subset of GCPs, and for each of the two models ($M_1$ and $M_2$), $J = 60$ perturbed calibrations were performed for analysis.

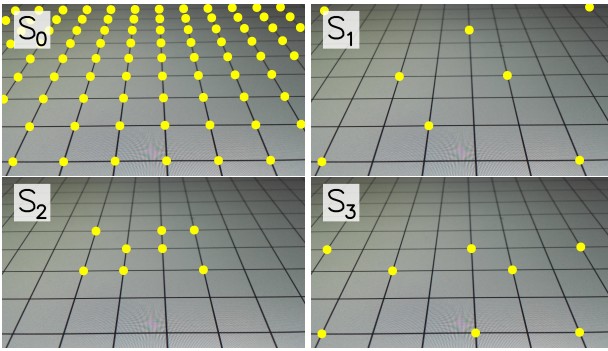

**Figure 4.** *Cont.*

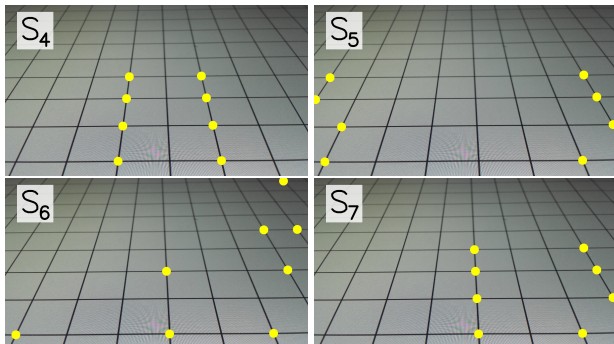

**Figure 4.** Subsets $S_0$ to $S_7$, for angle $A_1$, considered to analyze the influence of the GCPs distribution.

## 3. Results

The results for the two cameras, three angles, three series of number of GCPs, the eight GCP distributions, and for the two methods, are given in the Supplementary Materials. The main results are presented in this section.

### 3.1. Error Analysis

Figure 5 shows the distribution of the perturbed calibration errors $\epsilon_P$ for all the subsets of GCPs, for both models and for angle $A_1$ (the results for $A_2$ and $A_3$ were similar in this regard; not shown). Each boxplot contains information of the $J = 60$ perturbations. The calibration errors $\epsilon_P$ were smaller for $M_1$ than for $M_2$ for all subsets; this is a natural consequence of the model $M_2$ (with eight parameters) being a particular case of model $M_1$ (with 14 free parameters). However, it is noteworthy that model $M_2$, with around half of the parameters than $M_1$, still had small calibration errors, with $\epsilon_P \lesssim 3$ px. Also, from Figure 5, we can see that: (1) for $M_1$, the errors were larger for $S_0$ (i.e., using all the available points as GCPs); and (2) for $M_2$, the error was minimum for $S_2$ and $S_4$. Furthermore, there were no outliers; that is, all the calibrations can be considered to be satisfactory.

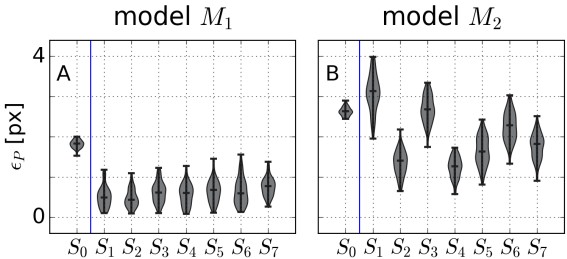

**Figure 5.** Errors $\epsilon_P$ for angle $A_1$ and models $M_1$ (**A**) and $M_2$ (**B**) as a function of the GCP calibration subset ($S_0$ to $S_7$).

As argued above, the error $\tilde{\epsilon}$ defined in Equation (5) gives us a better idea about the usability of the calibrations along the image. Figures 6 and 7 show the errors $\tilde{\epsilon}$ for all the available points for models $M_1$ and $M_2$, respectively, using the perturbed calibrations of the different subsets $S_k$ (the GCPs of the subsets $S_k$ are highlighted with small white circles, for ease of viewing). The results in Figures 6 and 7 are for the angle $A_1$ (the angles $A_2$ and $A_3$ showed the same trends, although with higher errors, as shown below through $\epsilon_Q$). As depicted in the figures, the errors remained small at the GCPs of each subset $S_k$. The behaviour outside the region $S_k$ was better for model $M_2$ than for $M_1$, especially for the cases $S_2$ and $S_4$–$S_7$; it can be seen that the color was saturated at $\tilde{\epsilon} = 20$ px, but the errors increased up to $\sim 10^3$ px for $S_2$ and $S_4$ in model $M_1$.

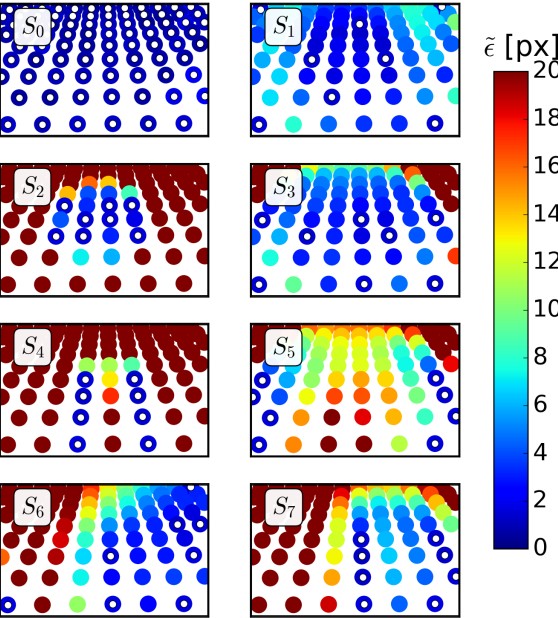

**Figure 6.** Errors $\tilde{\epsilon}$ for $M_1$ and angle $A_1$ at all the available points for the different sets $S_k$. The GCPs for each set are here highlighted with white circles in the center and correspond to the points in Figure 4.

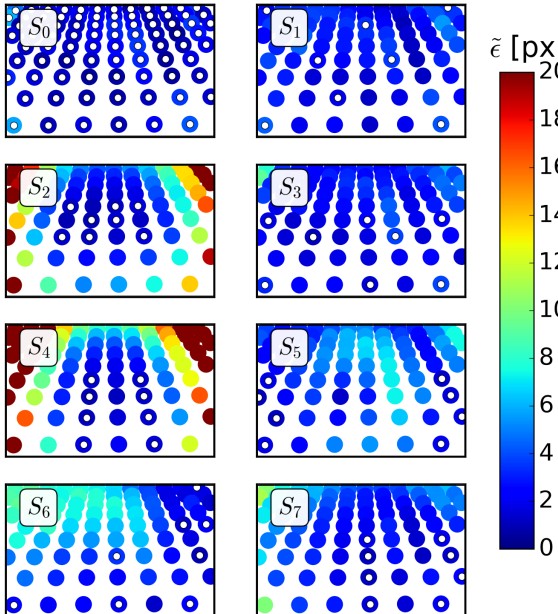

**Figure 7.** Errors $\tilde{\epsilon}$ for $M_2$ and angle $A_1$ at all the available points for the different sets $S_k$. The GCPs for each set are here highlighted with white circles in the center and correspond to the points in Figure 4.

Recalling Equation (6), Figure 8 shows the RMS of the errors $\tilde{\epsilon}$ in Figures 6 and 7 for angle $A_1$ and for all subsets $S_k$. The error $\epsilon_Q$ considers all the pixels in the image ($S_0$), while the error $\epsilon_G$ only considers the pixels used for the calibration (i.e., those highlighted in Figures 6 and 7) for the RMS. Naturally, both errors coincided for $S_0$. As already suggested from Figures 6 and 7, the errors $\epsilon_G$ were small in all cases; these errors were related to the errors $\epsilon_P$ in Figure 5. With regard to the error $\epsilon_Q$, which evaluates the quality of calibration in the whole image, model $M_2$ yielded significantly smaller errors than $M_1$, except for the very particular set $S_0$. For model $M_2$ (Figure 8B), all sets yielded overall errors $\epsilon_Q$ below 10, except for $S_2$ (pixels near the center of the image) and $S_4$ (centered in the lower half of the image). The sets $S_2$ and $S_4$ were those with smaller errors $\epsilon_P$ and $\epsilon_G$. The sets with smaller overall errors $\epsilon_Q$ were $S_1$ (ideal uniform distribution all over the image) and $S_3$ (lower half of the image), while sets $S_5$–$S_7$ gave similar results.

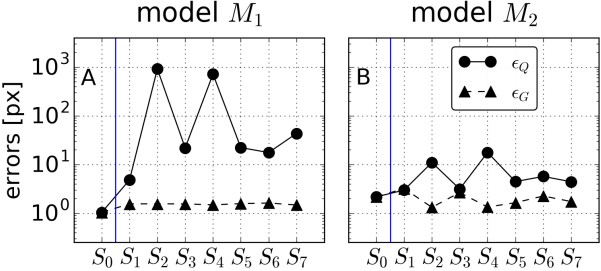

**Figure 8.** Errors $\epsilon_Q$ and $\epsilon_G$ for angle $A_1$ and models $M_1$ (**A**) and $M_2$ (**B**) as a function of the GCP calibration subset ($S_0$ to $S_7$).

### 3.2. Influence of the Obliquity of the Number of Gcps

The influence of the obliquity of the image on the errors $\epsilon_Q$ (as well as on $\epsilon_G$) is shown in Figure 9. This figure, an extension of Figure 8, includes the results for all three angles. The trends for angles $A_2$ and $A_3$ were, with respect to the model and the set $S_k$, similar to those described above for $A_1$. In particular, the errors $\epsilon_Q$ were, in general, too large for $M_1$ (despite the errors $\epsilon_G$ being very small). For model $M_2$, the errors $\epsilon_Q$ slightly increased for $A_2$ and $A_3$, subsets $S_2$ and $S_4$ giving the highest overall errors $\epsilon_Q$.

Similarly, the influence of the number of GCPs (for sets $S_1$ to $S_7$) on the errors $\epsilon_Q$ and $\epsilon_G$ is shown in Figure 10 for $A_1$. Figure 10 is an extension of Figure 8 and includes the results also for 6 and 12 GCPs. From Figure 10, it can be seen that the model $M_2$ keeps the overall errors $\epsilon_Q$ small, even with only 6 GCPs; except for $S_2$ and $S_4$. With regard to model $M_1$, while the errors $\epsilon_Q$ decreased for 12 GCPs (relative to 8 GCPs), they were still larger than for $M_2$. In the following section, we restrict again to 8 GCPs for $S_1$ to $S_7$.

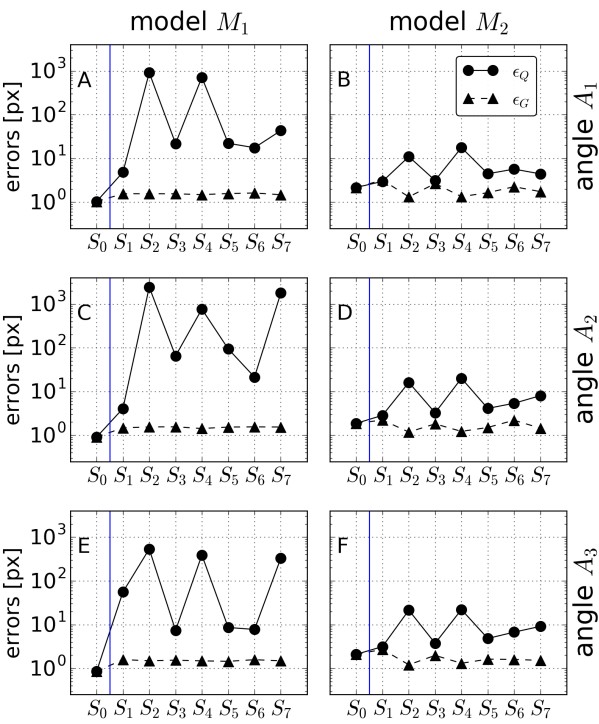

**Figure 9.** Errors $\epsilon_Q$ and $\epsilon_G$ for angle $A_1$ with $\tau \sim 55°$ (**A**,**B**); $A_2$ with $\tau \sim 40°$ (**C**,**D**); and $A_3$ with $\tau \sim 15°$ (**E**,**F**); and for models $M_1$ (**A**,**C**,**E**) and $M_2$ (**B**,**D**,**F**) as a function of the GCP calibration set ($S_0$ to $S_7$).

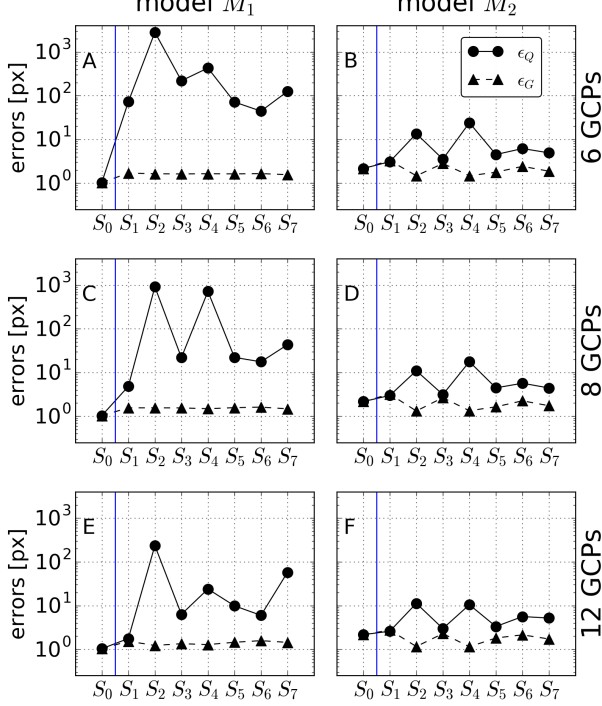

**Figure 10.** Errors $\epsilon_Q$ and $\epsilon_G$ for angle $A_1$ for different numbers of GCPs (for sets $S_1$ to $S_7$): 6 GCPs (**A**,**B**); 8 GCPs (**C**,**D**); and 12 GCPs (**E**,**F**) and for models $M_1$ (**A**,**C**,**E**) and $M_2$ (**B**,**D**,**F**).

### 3.3. Calibration Parameters

From a practical point of view, the above errors $\epsilon_Q$ are the most interesting results in the camera calibration problem. However, the recovery of the calibration parameters is also an issue of practical interest (e.g., recovering the camera position or the intrinsic parameters from a single snapshot). Figure 11 shows the results (using always all the $J$ perturbations) for the radial distortion $k_{1\star}$ and $s_{c\star}$ for both models and for angle $A_1$. Please note that the intrinsic parameters ($k_{1\star}$ and $s_{s\star}$ for $M_2$, and many other in the complete model $M_1$) must be independent of the angle considered– extrinsic parameters, on the contrary, depend on the image (angle)–. The information in this figure contains the results for $A_1$, the results fro $A_2$ and $A_3$ being similar (not shown). From Figure 11, the results for $M_1$ show a large variability when compared to those for model $M_2$. Model $M_2$, except for sets $S_2$ and $S_4$ –and in particular for the radial distortion $k_{1\star}$–, shows small standard deviations in the boxplots. Having small standard deviations means that all perturbed calibrations give similar values of the parameters, so that the results are trustable. The rest of intrinsic parameters in model $M_1$ ($k_{2\star}$, $p_{1\star}$, …) have a similar behaviour than that of $k_{1\star}$ and $s_{c\star}$ (i.e., with large standard deviations, not shown).

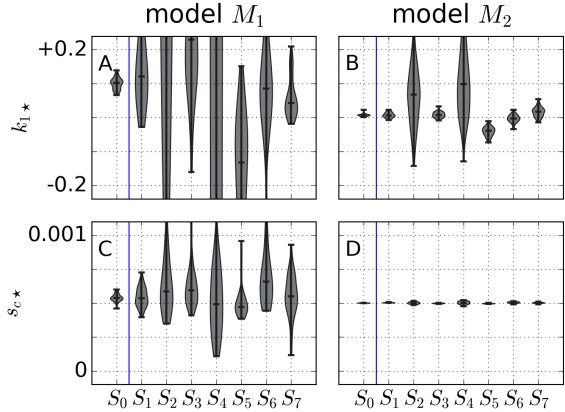

**Figure 11.** Radial distortion $k_{1\star}$ (**A,B**) and pixel size $s_{c\star}$ (**C,D**) for models $M_1$ (**A,C**) and $M_2$ (**B,D**) for angle $A_1$.

Given that the model $M_2$ performed similar to $M_1$ in terms of $\epsilon_G$, while giving smaller overall errors $\epsilon_Q$ (Figure 8) and, further, provides more trustable results for the intrinsic parameters, we will focus on $M_2$ for the extrinsic parameters (model $M_1$ provides noisy results for the extrinsic parameters, as it does for the intrinsic ones; not shown).

The extrinsic parameters ($x_c$, $y_c$, $z_c$, $\phi$, $\sigma$, and $\tau$) depend on the image (angle) considered, as already mentioned. Figure 12 shows the results for the camera position ($x_c$, $y_c$, and $z_c$) for angles $A_1$–$A_3$ using the reduced model $M_2$. For each angle, given that the results for $S_0$ (with ∼80 GCPs) had the smallest standard deviation (i.e., were the most trustable), the mean value for $S_0$ was subtracted in all cases ($\overline{x_{c,S_0}}$, $\overline{y_{c,S_0}}$, $\overline{z_{c,S_0}}$). In this way, the variability of the parameter is shown for each angle $A_i$ independently of the actual values of the parameters, which are of minor interest here (and different for all three cases). From Figure 12, angle $A_1$ (with the larger obliquity) produced good estimates of the camera position, except (again) when using sets $S_2$ and $S_4$. The results worsened for angles $A_2$ and, especially, $A_3$ (∼zenithal). The results for the Eulerian angles $\phi$, $\sigma$, and $\tau$ are shown in Figure 13. The results followed the same trends as for the camera position; that is, case $A_1$ gave more robust results than $A_2$ and much more than $A_3$, and $S_2$ and $S_4$ performed especially bad. It is worth noting that $\overline{\tau_{S_0}} = 0.95$ rad $\approx 54°$ for $A_1$, $\overline{\tau_{S_0}} = 0.76$ rad $\approx 44°$ for $A_2$, and $\overline{\tau_{S_0}} = 0.37$ rad $\approx 21°$ for $A_3$.

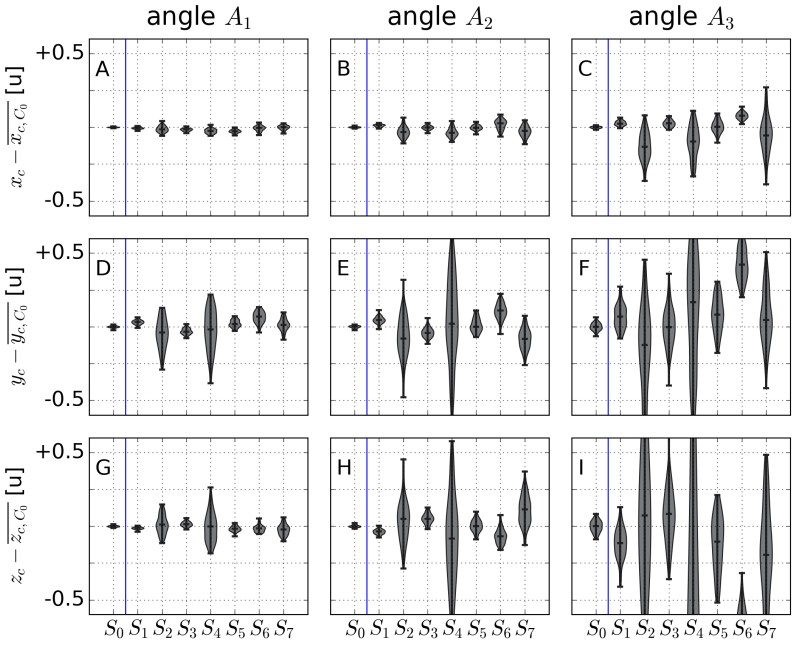

**Figure 12.** Demeaned camera position co-ordinates $x_c$, $y_c$, and $z_c$ for angles $A_1$ (**A,D,G**), $A_2$ (**B,E,H**), and $A_3$ (**C,F,I**) for model $M_2$. The unit length "u" corresponds to the side of the squares of the grid.

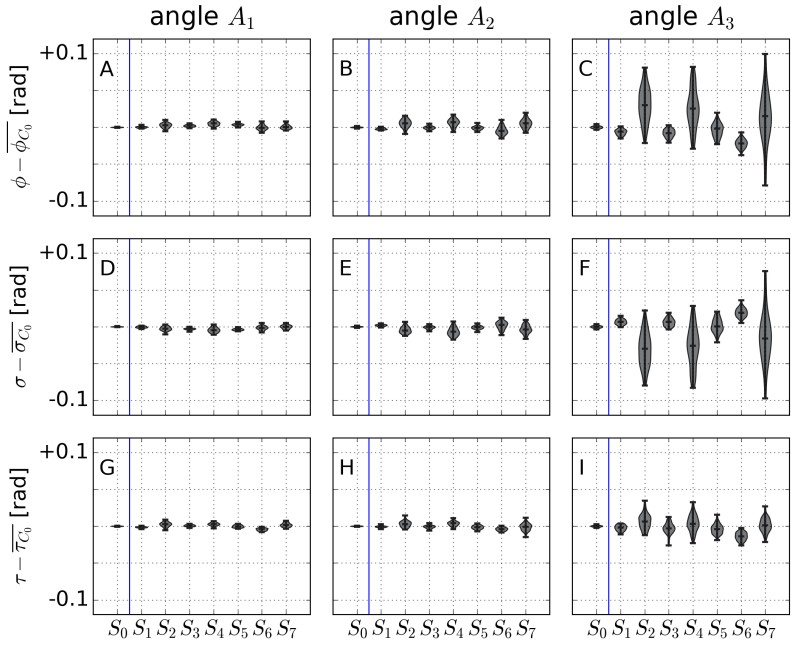

**Figure 13.** Demeaned camera Eulerian angles $\phi$, $\sigma$, and $\tau$ for angles $A_1$ (**A,D,G**), $A_2$ (**B,E,H**), and $A_3$ (**C,F,I**) for model $M_2$.

## 4. Discussion

The above results—on full calibration of a camera from one single snapshot—show that there is no correlation of the overall quality of the calibration (which can be measured in terms of $\epsilon_Q$) with the error obtained in the optimization process to obtain the calibration parameters. However, in real calibrations, the error $\epsilon_Q$ cannot be known, while only $\epsilon^*$ (similar to $\epsilon_P$ and $\epsilon_G$) can be obtained.

The latter errors being small only ensures, in general, good performance of the calibration around the calibration GCPs (Figures 6 and 7 are clear, in this regard).

The results show that the choice of GCPs is crucial to obtain an effective real calibration (i.e., minimal $\epsilon_Q$ values). Ideally, the overall calibration errors $\epsilon_Q$ should be minimized by using a large number of GCPs covering the entire image. However, in real situations, the calibration GCPs are limited to a small region of the image, while other parts of the image can be of interest to the research. For example, in Figure 1B, the GCPs would usually be located in the promenade (where there are lots of observable features), while the focus is on the shoreline or the water area. Furthermore, the amount of GCPs is limited for functional requirements. Our findings show that good quality calibrations can be obtained with a limited number of GCPs when at least some of them are placed at the edges of the image. In these cases, even without having the smallest $\epsilon_G$, the $\epsilon_Q$ errors are small. On the other hand, when all the GCPs are centered in the image, the calibration quality may be poor (large $\epsilon_Q$), even if $\epsilon_G$ are small. The justification for this and other behaviours is presented below.

The selection of an appropriate calibration model is essential. Ideally, when a large number of GCPs are available and cover the whole image, the complete model ($M_1$) is the best, both with regard to $\epsilon_G$ and $\epsilon_Q$ (Figure 8 for $S_0$). This can typically be done under laboratory conditions but is not the case in coastal studies; particularly when taking advantage of freely available coastal images. For a realistic set of GCPs, the reduced model $M_2$ provided, in all studied cases, the highest quality calibrations. Again, we found the (kind of) paradoxical result that the best $\epsilon_Q$ were obtained with the model $M_2$, although the calibration errors were always smaller in model $M_1$ and, therefore, could seem to be more robust. From the above results (Figure 10), the advantage of the model $M_2$ compared to $M_1$ is evident for a reduced number of GCPs (6), remaining even when it is incremented to more reasonable values (12).

The model $M_2$ behaving better than $M_1$ is related to the noise in the recovery of the calibration parameters for model $M_1$ (illustrated in Figure 11 for $k_{1\star}$ and $s_{c\star}$), as explained below. Having just *one* snapshot to perform the calibration may lead, especially if the GCP distribution is not favourable (as in $S_2$ or $S_4$), to many different combinations of parameters providing small calibration errors ($\epsilon^* \sim \epsilon_G$) but large overall errors $\epsilon_Q$. In the complete model, $M_1$, this compensation of different calibration variables to give similar calibration errors $\epsilon^*$ is much more pronounced, as it contains more parameters: this explains the large deviations of the parameters $k_{1\star}$ and $s_{c\star}$ in Figure 11 (and also in the rest of the calibration parameters; not shown) and the larger errors $\epsilon_Q$, except for in $S_0$ (Figures 9 and 10). Model $M_1$ was overparametrized for 6 GCPs and, for 8 and 12 GCPs, still showed symptoms of overparametization behaviours. Focusing on the simple model, $M_2$, the above compensation mechanism shows up in the worse case $S_2$ (and in $S_4$). In the model $M_2$, the role of the physical distance from the camera position to the GCPs (i.e., the co-ordinates of $\mathbf{x_c}$), the size $s_{c\star}$ and the distortion $k_{1\star}$ can be compensated if the GCPs fall near the center of the image, when the role of the distortion cannot be clearly distinguished. This the reason the set $S_2$ showed large deviations in the camera position (see Figure 12) and $k_{1\star}$ (Figure 11 for $M_2$). For this model, these mechanisms were enhanced for small $\tau$ (angle $A_3$, Figures 12 and 13), giving slightly larger errors $\epsilon_Q$ in Figure 9. The angle $A_1$ gave more robust results (in the calibration parameters) due to the fact that, by increasing the relative distances between the different GCPs, the calibration parameters were more accurately captured. Zenithal images with the GCPs concentrated in the center of the image led to the worst quality calibration errors $\epsilon_Q$, despite achieving an excellent calibration error $\epsilon_G$ (Figure 9, set $S_2$).

For calibration purposes, we recommend the use of model $M_2$ and the selection of the GCPs such that some of them fall near the edges of the image. Whenever the recovery of the camera position and orientation is of interest, using zenithal views should be avoided. The use of the simple model $M_2$ to properly georeference images obtained by different devices using just a few GCPs opens up a range of possibilities for the analysis of images from webcams or beach users and the quantification of different parameters of interest (e.g., position and shape of the coastline, ...). Furthermore, in fixed video monitoring systems, even if the camera has been intrinsically calibrated prior to its final deployment,

the intrinsic calibration (as well as the extrinsic one) can change in time, due to changing external conditions, and continuous re-calibration of the parameters may be required.

## 5. Conclusions

In this work, we analyzed the influence that the distribution of GCPs and image obliquity has on the overall quality of full (intrinsic and extrinsic) camera calibration using only a single snapshot. This was done by analyzing the performance of two calibration mathematical models. We conclude that, for the calibration of coastal images—especially when only one image is available—the reduced model should be used. This reduced model provided robust camera calibration parameters (camera position, Eulerian angles, pixel size, and radial distortion) in our tests, allowing for an explicit transformation from pixel to real-world co-ordinates and, most importantly, yielded smaller overall calibration errors. With respect to the distribution of the GCPs over the image, using calibration points only near the centre of the image must be avoided, and we recommend using the maximum number of points distributed along the edges of the image. Finally, zenithal views complicate the recovery of the calibration parameters, although the obliquity does not have a significant influence on the overall performance of the calibration.

**Supplementary Materials:** The following are available online at http://www.mdpi.com/2072-4292/12/11/1840/s1.

**Author Contributions:** Conceptualization, G.S., D.C., P.S., and J.G.; methodology, G.S. and D.C.; software, G.S. and D.C.; validation, G.S., D.C., and P.S.; formal analysis, G.S. and D.C.; investigation, G.S., D.C., P.S., and J.G.; resources, G.S., D.C., and J.G.; writing–original draft preparation, G.S. and D.C.; writing–review and editing, G.S., D.C., P.S., and J.G.; visualization, G.S. and D.C.; supervision, G.S., D.C., P.S., and J.G.; project administration, G.S. and D.C.; funding acquisition, G.S., D.C., and J.G. All authors have read and agreed to the published version of the manuscript.

**Funding:** This research was funded by the Spanish Government (MINECO/MICINN/FEDER) grant numbers RTI2018-093941-B-C32, and RTI2018-093941-B-C33.

**Acknowledgments:** The authors also acknowledge the useful suggestions from R. Álvarez and G. Grande.

**Conflicts of Interest:** The authors declare no conflict of interest.

## Abbreviations

The following abbreviations are used in this manuscript:

CPU      Central Processing Unit
GCP(s)    Ground Control Point(s)
RMS      Root Mean Square

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
