# Peer review of "Camera Calibration for Coastal Monitoring Using Available Snapshot Images"

_remotesensing, doi:10.3390/rs12111840_

Round 1

Reviewer 1 Report

"The present paper is on the topic of camera calibration. The paper is well written, while this topic has beeen widely treated in literature and more references should be discussed in the introduction.

In fact "the introduction should briefly place the study in a broad context and highlight why it is important. It should define the purpose of the work and its significance, including specific hypotheses being tested. The current state of the research field should be reviewed carefully and key publications cited. Please highlight controversial and diverging hypotheses when necessary. Finally, briefly mention the main aim of the workand highlight the main conclusions. Keep the introduction comprehensible to scientists working outside the topic of the paper."

As a consequence the authors should discuss deeply the difference between calibrating in very close, close and wide angle field of view.

Please refer to:

Close range:

Ricolfe-Viala, C., Sánchez-Salmerón, A.-J. Robust metric calibration of non-linear camera lens distortion (2010) Pattern Recognition, 43 (4), pp. 1688-1699.

Very close range:

Papers dealing with close-range camera calibration model for narrow angles of view.

Moreover, in the introduction the authors should discuss the difference between single camera and multi camera approach. As regards multicamera you to several papers dealing, for instance, with the photogrammetric acquisition of the soft tissues of the face.

One example can be: Galantucci, L.M., Ferrandes, R., Percoco, G.Digital photogrammetry for facial recognition(2006) Journal of Computing and Information Science in Engineering, 6 (4), pp. 390-396.

One very important concern regards that the authors clarify very well their contribution to the topic."

Author Response

First of all, we thank the Reviewer for the effort to revise the document as well as for his/her constructive comments. We have considered all your comments to improve the manuscript. You will find a detailed reply below. We have upload a version of the manuscript with the track of modifications. Changes motivated by your comments are indicated with "R1" (REVIEWER 1). Please, note that all the changes related to this Reviewer correspond to the introduction and in the section 2.1 (camera mathematical models).

Point 1: In fact "the introduction should briefly place the study in a broad context and highlight why it is important. It should define the purpose of the work and its significance, including specific hypotheses being tested.

Response 1: We did not place the study in a broader context following the usual standards in works related to camera calibration for coastal video monitoring. However, we understand that it is worth placing the work in a broad context. Actually, even though the work is motivated in the coastal monitoring field, we hope that the results can be of interest for a wider community. We have modified of the manuscript (mainly the introduction) following this Reviewer comments.

Point 2: The current state of the research field should be reviewed carefully and key publications cited. Please highlight controversial and diverging hypotheses when necessary.

Response 2: We have added some relevant references. Besides the ones suggested by the Reviewer, we added classical references and some other relevant more recent references (e.g., Brown '71, Zhang '00 and Zhang '18 among other). We also mentioned the assumptions of the models.

Point 3: Finally, briefly mention the main aim of the work and highlight the main conclusions.

Response 3: Following the Reviewer suggestions, at the end of the introduction we now state more clearly the assumptions and the goal of the manuscript, in accordance with the conclusions of the paper.

Point 4: Keep the introduction comprehensible to scientists working outside the topic of the paper." As a consequence the authors should discuss deeply the difference between calibrating in very close, close and wide angle field of view.

Response 4: We have broaden the introduction to put the work more into context and make it more appealing for the community working on camera calibration and not only in coastal video monitoring. However, we prefer not to extend it with too many details in regard different models, methods and their characteristics. Just as an example, there is a lot of interesting literature on DLT and related methods (including the recent work by Zhang'18) whose details are avoided. We consider a well known nonlinear lens distortion / decentering model and the parameters are found by minimizing the reprojection error.

Point 5: Please refer to: Close range: Ricolfe-Viala, C., Sánchez-Salmerón, A.-J. Robust metric calibration of non-linear camera lens distortion (2010) Pattern Recognition, 43 (4), pp. 1688-1699. Very close range: Papers dealing with close-range camera calibration model for narrow angles of view.

Response 5: We have included this reference (and other).

Point 6: Moreover, in the introduction the authors should discuss the difference between single camera and multi camera approach. As regards multicamera you to several papers dealing, for instance, with the photogrammetric acquisition of the soft tissues of the face. One example can be: Galantucci, L.M., Ferrandes, R., Percoco, G.Digital photogrammetry for facial recognition(2006) Journal of Computing and Information Science in Engineering, 6 (4), pp. 390-396.

Response 6: Following the suggestion, we have mentioned the topic in the introduction. However, since the multi camera approach is mainly used in coastal monitoring to determine wave characteristics, and this is out of the focus of this work, we preferred not to go into details.

Point 7: One very important concern regards that the authors clarify very well their contribution to the topic.

Response 7: Following the Reviewer suggestion, we have clearly emphasized what is the contribution of the work (in the introduction): the goal is on camera calibration with 1 image and "few" GCPs, analyzing the influence of GCPs distribution and mathematical model.

Reviewer 2 Report

It is a good work, complete of everything. Only, English must be deeply reconsidered.

Author Response

We are pleased with the positive reception of the work.

Point 1: It is a good work, complete of everything. Only, English must be deeply reconsidered.

Response 1: Following the suggestion of the Reviewer, we have improved the English of the manuscript by sending it to be proofread by the professional MDPI English editing service.

Reviewer 3 Report

A quite interesting manuscript. In my humble opinion it only lacks some more clarification in some aspects (a new sketch, some aditional explanations, ...) which have been highlighted in the attached file

Author Response

We thank the Reviewer for the effort to revise the document as well as for his/her constructive comments. We are pleased with the positive reception of the work. We have considered all your comments to improve the manuscript. This includes also a new figure and "Complementary Material" that includes results for the two cameras and all the experiment cases. You will find a detailed reply below. All lines / figures correspond to the previous version. We have upload a version of the manuscript with the track of modifications. Changes motivated by your comments are indicated with "R3" (REVIEWER 3). For your ease, we have also uploaded a modification of your pdf including the replies.

Point 1: A quite interesting manuscript. In my humble opinion it only lacks some more clarification in some aspects (a new sketch, some additional explanations, ...) which have been highlighted in the attached file.

Response 1: Following the Reviewer suggestion, we have added the required sketch as well as additional explanations.

line 36: An explanation about why not to locate the GCPs on the sand would be included.

Response: We have included an explanation in the new version of the manuscript. During a calibration campaign, GCPs can be chosen wherever required by using targets; even in sand and water. However, when calibrating an image taken at a different time, eg. images provided by other users, it is only possible to use fixed features, thereby excluding the regions with sand and water.

line 55: I guess that a sketch is here necessary to clarify all these concepts.

Response: Following the Reviewer's suggestion, we have added a sketch that illustrates some of the calibration parameters, particularly the camera angles and positions. We expect it to be more clarifying for the reader. Thanks to the reviewer's comment, we have found a typo (tilt<>roll) that has also been corrected.

line 75: Is this procedure based on a former one? Please, provide a cite.

Response: Yes, it is actually the same procedure introduced by Simarro et al. (2017) --matlab calibrator uses the same goal function--. Following the suggestion, we have added the reference also here (it was only in the introduction).

line 91: Pixels, I guess.

Response: Yes indeed. We have modified the manuscript.

line 94: What is the error in the determination of these coordinates?
What was the method used?

Response: In regard the real-world coordinates, their error is negligible in coastal studies (tens of millimeters usually, much less, in any case, than the size that the pixel represents in the real world) and it was also negligible in the grid considered in this study (created with python and displayed on a laptop screen). We have neglected them in this study. We have added a short comment in this regard in the new version of the manuscript.

line 102: Root mean square error? not explained before.

Response: We are sorry about it. We have modified the manuscript to introduce it here.

line 105: Please, clarify.

Response: We have added an explanation of why such equality is achieved.

line 112: Sorry, 3 or 4 snapshots?

Response: They are 3 snapshots indeed. We have rewritten the manuscript at this point (stating that tau = 0 is the zenithal view was probably misleading).

line 115: Would the authors please note a relationship between px units and cm in this case?

Response: The images are of 2048 x 1512 pixels, and the largest squares are, in the real-world, about 5cm (the unit length "u"). In the new version of the manuscript we have stated that u is approximately 5cm. We do not think that this is actually relevant, but we understand that giving the value can help the reader.

caption of Figure 2: Please, specify.

Response: Following the suggestion, we have included the approximated values of the angles in the caption of the Figure.

line 125: I am not quite sure of that.

Response: While in laboratory conditions the number of GCPs is much higher, in many coastal video monitoring stations that can be a reasonable number. Nieto et al. (2010) suggest 8 GCPs while Harley et al. (2019, CoastSnap) consider 7 points in their method 1. Please note that we also show the results for 6 and 12 GCPs. We have added the CoastSnap reference here.

line 133: Why name it into the Methodology, then?

Response: The results for A2 and A3 "in regard epsilon_P" are similar and not shown. The results for A2 and A3 are actually shown in the results (and discussed) in other regards. We have modified the manuscript to make it clear that are similar "in this regard". Also, we have attached a complementary file that includes the plots for all the cases. Now the results section starts by reporting that all the results are in the Supplementary Material. In the results section, the main results are presented.

caption of Figure 5: Is there any possibility of a graphic relation between the errors shown here and the subsets of figure 3?

Response: Actually, the highlighted white dots in Figure 5 correspond to the points in Figure 3. We already made a comment in the caption of Figure 5 in this regard. We have emphasized it in the caption (making a reference to Figure 3).

line 161: As the results for the tree angles are eventually shown, a small sentence should be included in lines 133 and 145 to guide the reader.

Response: Yes. We have done it in line 133 (always in the old manuscript) and also in line 145. Now, results for all angles are shown at the Supplementary Material.

line 169: Please, check if an explanation has been provided in the discussion section.

Response: Yes. We checked that this point is treated in the discussion. Actually, we have added a new line (also related to the last comment of the Reviewer) in the new version of the manuscript.

caption of Figure 8: Please, check size and colour of angle A1.

Response: It was actually done on purpose, so as to emphasize the fact that A1 was the case shown in Figure 7. However, following the suggestion, we changed the size and colour so that all three cases look the same.

caption of Figure 8: Please, identify the angles at the figure caption.

Response: We have included the angles in the caption following the suggestion.

line 188: Excuse me, why not?

Response: We prefer not to introduce them for they are very large and we think that they could mislead the reader (for its size and little amount of information). Following this Reviewer suggestion, however, we have created a file with additional material where we show all the plots generated (all cases: 2 cameras, 3 angles, 3 numbers of GCPs, 2 methods): we pretend this file to be accessible to the reader.

caption of Figure 9: Check size and colour for 8 GCPs.

Response: As above, it was actually done on purpose, so as to emphasize the fact that 8GCPs was the case shown in Figure 7. However, following the suggestion, we changed the size and colour so that all three cases look the same.

caption of Figure 10:
Please, clarify.

Response: The pixel size sc* is defined in the introduction.

line 214: This is a very interesting conclusion, but in my humble opinion it should be reinforced with a complementary justification.

Response: Please, see reply to the next comment.

line 225: I am afraid that this reviewer do really need an easier explanation which explained why M2 yields smaller overall calibration errors.

Response: The reasons are explained in the following paragraph. In summary, for few GCPs, model M1 (with 14 free parameters) seems to be overparametrized. On top of that, the GCPs not being well distributed can make the calibration through model M1 useless in practice. Taking into account the Reviewer comment, we have slightly modified the manuscript at that paragraph trying to be more clear.

Round 2

Reviewer 1 Report

The paper can be published in its present form

Reviewer 2 Report

OK

Reviewer 3 Report

Nice work.

Congratulations